# Estimation with Norm Regularization

**Arindam Banerjee**    **Sheng Chen**    **Farideh Fazayeli**    **Vidyashankar Sivakumar**

Department of Computer Science & Engineering
University of Minnesota, Twin Cities
{banerjee,shengc,farideh,sivakuma}@cs.umn.edu

## Abstract

Analysis of non-asymptotic estimation error and structured statistical recovery based on norm regularized regression, such as Lasso, needs to consider four aspects: the norm, the loss function, the design matrix, and the noise model. This paper presents generalizations of such estimation error analysis on all four aspects. We characterize the restricted error set, establish relations between error sets for the constrained and regularized problems, and present an estimation error bound applicable to *any* norm. Precise characterizations of the bound is presented for a variety of noise models, design matrices, including sub-Gaussian, anisotropic, and dependent samples, and loss functions, including least squares and generalized linear models. Gaussian width, a geometric measure of size of sets, and associated tools play a key role in our generalized analysis.

## 1 Introduction

Over the past decade, progress has been made in developing non-asymptotic bounds on the estimation error of structured parameters based on norm regularized regression. Such estimators are usually of the form [16, 9, 3]:

$$\hat{\theta}_{\lambda_n} = \operatorname*{argmin}_{\theta \in \mathbb{R}^p} \mathcal{L}(\theta; Z^n) + \lambda_n R(\theta) \,, \tag{1}$$

where $R(\theta)$ is a suitable norm, $\mathcal{L}(\cdot)$ is a suitable loss function, $Z^n = \{(y_i, X_i)\}_{i=1}^n$ where $y_i \in \mathbb{R}$, $X_i \in \mathbb{R}^p$ is the training set, and $\lambda_n > 0$ is a regularization parameter. The optimal parameter $\theta^*$ is often assumed to be 'structured,' usually characterized as low value according to some norm $R(\cdot)$. Since $\hat{\theta}_{\lambda_n}$ is an estimate of the optimal structure $\theta^*$, the focus has been on bounding a suitable function of the error vector $\hat{\Delta}_n = (\hat{\theta}_{\lambda_n} - \theta^*)$, e.g., the $L_2$ norm $\|\hat{\Delta}_n\|_2$.

To understand the state-of-the-art on non-asymptotic bounds on the estimation error for norm-regularized regression, four aspects of (1) need to be considered: (i) the norm $R(\theta)$, (ii) properties of the design matrix $X \in \mathbb{R}^{n \times p}$, (iii) the loss function $\mathcal{L}(\cdot)$, and (iv) the noise model, typically in terms of $w = y - E[y|x]$. Most of the literature has focused on a linear model: $y = X\theta + \omega$, and a squared-loss function: $\mathcal{L}(\theta; Z^n) = \frac{1}{n}\|y - X\theta\|_2^2 = \frac{1}{n}\sum_{i=1}^n (y_i - \langle \theta, X_i \rangle)^2$. Early work on such estimators focussed on the $L_1$ norm [21, 20, 8], and led to sufficient conditions on the design matrix $X$, including the restricted-isometry properties (RIP) and restricted eigenvalue (RE) conditions [2, 9, 13, 3]. While much of the development has focussed on isotropic Gaussian design matrices, recent work has extended the analysis for $L_1$ norm to correlated Gaussian designs [13] as well as anisotropic sub-Gaussian design matrices [14].

Building on such development, [9] presents a unified framework for the case of decomposable norms and also considers generalized linear models (GLMs) for certain norms such as $L_1$. Two key insights are offered in [9]: first, for suitably large $\lambda_n$, the error vector $\hat{\Delta}_n$ lies in a restricted set, a cone or a star, and second, on the restricted error set, the loss function needs to satisfy restricted strong convexity (RSC), a generalization of the RE condition, for the analysis to work out.

For isotropic Gaussian design matrices, additional progress has been made. [4] considers a constrained estimation formulation for all atomic norms, where the gain condition, equivalent to the RE condition, uses Gordons inequality [5, 7] and is succinctly represented in terms of the Gaussian width of the intersection of the cone of the error set and a unit ball/sphere. [11] considers three related formulations for generalized Lasso problems, establish recovery guarantees based on Gordons inequality, and quantities related to the Gaussian width. Sharper analysis for recovery has been considered in [1], yielding a precise characterization of phase transition behavior using quantities related to the Gaussian width. [12] consider a linear programming estimator in a 1-bit compressed sensing setting and, interestingly, the concept of Gaussian width shows up in the analysis. In spite of the advances, most of these results are restricted to isotropic Gaussian design matrices.

In this paper, we consider structured estimation problems with norm regularization, which substantially generalize existing results on all four pertinent aspects: the norm, the design matrix, the loss, and the noise model. The analysis we present applies to *all* norms. We characterize the structure of the error set for all norms, develop precise relationships between the error sets of the regularized and constrained versions [2], and establish an estimation error bound in Section 2. The bound depends on the regularization parameter $\lambda_n$ and a certain RSC condition constant $\kappa$. In Section 3, for both Gaussian and sub-Gaussian noise $\omega$, we develop suitable characterizations for $\lambda_n$ in terms of the Gaussian width of the unit norm ball $\Omega_R = \{u|R(u) \le 1\}$. In Section 4, we characterize the RSC condition for any norm, considering two families of design matrices $X \in \mathbb{R}^{n \times p}$: Gaussian and sub-Gaussian, and three settings for each family: independent isotropic designs, independent anisotropic designs where the rows are correlated as $\Sigma_{p \times p}$, and dependent isotropic designs where the rows are isotropic but columns are correlated as $\Gamma_{n \times n}$, implying dependent samples. In Section 5, we show how to extend the analysis to generalized linear models (GLMs) with sub-Gaussian design matrices and any norm.

Our analysis techniques are simple and largely uniform across different types of noise and design matrices. Parts of our analysis are geometric, where Gaussian widths, as a measure of size of suitable sets, and associated tools play a key role [4, 7]. We also use standard covering arguments, use Sudakov-Dudley inequality to switch from covering numbers to Gaussian widths [7], and use generic chaining to upper bound 'sub-Gaussian widths' with Gaussian widths [15].

## 2 Restricted Error Set and Recovery Guarantees

In this section, we give a characterization of the restricted error set $E_r$ in which the error vector $\hat{\Delta}_n$ lives, establish clear relationships between the error sets for the regularized and constrained problems, and finally establish upper bounds on the estimation error. The error bound is deterministic, but has quantities which involve $\theta^*, X, \omega$, for which we develop high probability bounds in Sections 3, 4, and 5.

### 2.1 The Restricted Error Set and the Error Cone

We start with a characterization of the restricted error set $E_r$ where $\hat{\Delta}_n$ will belong.

**Lemma 1** *For any $\beta > 1$, assuming*

$$\lambda_n \ge \beta R^*(\nabla \mathcal{L}(\theta^*; Z^n)) \,, \tag{2}$$

*the error vector $\hat{\Delta}_n = \hat{\theta}_{\lambda_n} - \theta^*$ belongs to the set*

$$E_r = E_r(\theta^*, \beta) = \left\{ \Delta \in \mathbb{R}^p \,\middle|\, R(\theta^* + \Delta) \le R(\theta^*) + \frac{1}{\beta} R(\Delta) \right\} \,. \tag{3}$$

The restricted error set $E_r$ need not be convex for general norms. Interestingly, for $\beta = 1$, the inequality in (3) is just the triangle inequality, and is satisfied by all $\Delta$. Note that $\beta > 1$ restricts the set of $\Delta$ which satisfy the inequality, yielding the restricted error set. In particular, $\Delta$ cannot go in the direction of $\theta^*$, i.e., $\Delta \ne \alpha \theta^*$ for any $\alpha > 0$. Further, note that the condition in (2) is similar to that in [9] for $\beta = 2$, but the above characterization holds for any norm, not just decomposable norms [9].

While $E_r$ need not be a convex set, we establish a relationship between $E_r$ and $C_c$, the cone for the constrained problem [4], where

$$C_c = C_c(\theta^*) = \text{cone} \{\Delta \in \mathbb{R}^p \mid R(\theta^* + \Delta) \leq R(\theta^*)\} \ . \tag{4}$$

**Theorem 1** *Let $A_r = E_r \cap \rho B_2^p$ and $A_c = C_c \cap \rho B_2^p$, where $B_2^p = \{u \mid \|u\|_2 \leq 1\}$ is the unit ball of $\ell_2$ norm and $\rho > 0$ is any suitable radius. Then, for any $\beta > 1$ we have*

$$w(A_r) \leq \left(1 + \frac{2}{\beta - 1} \frac{\|\theta^*\|_2}{\rho}\right) w(A_c) \ , \tag{5}$$

*where $w(A)$ denotes the Gaussian width of any set $A$ given by: $w(A) = E_g[\sup_{a \in A} \langle a, g \rangle]$, where $g$ is an isotropic Gaussian random vector.*

Thus, the Gaussian width of the error sets of regularized and constrained problems are closely related. In particular, for $\|\theta^*\|_2 = 1$, with $\rho = 1, \beta = 2$, we have $w(A_r) \leq 3w(A_c)$. Related observations have been made for the special case of the $L_1$ norm [2], although past work did not provide an explicit characterization in terms of Gaussian widths. The result also suggests that it is possible to move between the error analysis of the regularized and the constrained versions of the estimation problem.

## 2.2 Recovery Guarantees

In order to establish recovery guarantees, we start by assuming that restricted strong convexity (RSC) is satisfied by the loss function in $C_r = \text{cone}(E_r)$, i.e., for any $\Delta \in C_r$, there exists a suitable constant $\kappa$ so that

$$\delta\mathcal{L}(\Delta, \theta^*) \triangleq \mathcal{L}(\theta^* + \Delta) - \mathcal{L}(\theta^*) - \langle \nabla \mathcal{L}(\theta^*), \Delta \rangle \geq \kappa \|\Delta\|_2^2 \ . \tag{6}$$

In Sections 4 and 5, we establish precise forms of the RSC condition for a wide variety of design matrices and loss functions. In order to establish recovery guarantees, we focus on the quantity

$$\mathcal{F}(\Delta) = \mathcal{L}(\theta^* + \Delta) - \mathcal{L}(\theta^*) + \lambda_n(R(\theta^* + \Delta) - R(\theta^*)) \ . \tag{7}$$

Since $\hat{\theta}_{\lambda_n} = \theta^* + \hat{\Delta}_n$ is the estimated parameter, i.e., $\hat{\theta}_{\lambda_n}$ is the minimum of the objective, we clearly have $\mathcal{F}(\hat{\Delta}_n) \leq 0$, which implies a bound on $\|\hat{\Delta}_n\|_2$. Unlike previous results, the bound can be established without making any additional assumptions on the norm $R(\theta)$. We start with the following result, which expresses the upper bound on $\|\hat{\Delta}_n\|_2$ in terms of the gradient of the objective at $\theta^*$.

**Lemma 2** *Assume that the RSC condition is satisfied in $C_r$ by the loss $\mathcal{L}(\cdot)$ with parameter $\kappa$. With $\hat{\Delta}_n = \hat{\theta}_{\lambda_n} - \theta^*$, for any norm $R(\cdot)$, we have*

$$\|\hat{\Delta}_n\|_2 \leq \frac{1}{\kappa}\|\nabla\mathcal{L}(\theta^*) + \lambda_n \nabla R(\theta^*)\|_2 \ , \tag{8}$$

*where $\nabla R(\cdot)$ is any sub-gradient of the norm $R(\cdot)$.*

Note that the right hand side is simply the $L_2$ norm of the gradient of the objective evaluated at $\theta^*$. For the special case when $\hat{\theta}_{\lambda_n} = \theta^*$, the gradient of the objective is zero, implying correctly that $\|\hat{\Delta}_n\|_2 = 0$. While the above result provides useful insights about the bound on $\|\hat{\Delta}_n\|_2$, the quantities on the right hand side depend on $\theta^*$, which is unknown. We present another form of the result in terms of quantities such as $\lambda_n$, $\kappa$, and the norm compatibility constant $\Psi(C_r) = \sup_{\mathbf{u} \in C_r} \frac{R(\mathbf{u})}{\|\mathbf{u}\|_2}$, which are often easier to compute or bound.

**Theorem 2** *Assume that the RSC condition is satisfied in $C_r$ by the loss $\mathcal{L}(\cdot)$ with parameter $\kappa$. With $\hat{\Delta}_n = \hat{\theta}_{\lambda_n} - \theta^*$, for any norm $R(\cdot)$, we have*

$$\|\hat{\Delta}_n\|_2 \leq \frac{1 + \beta}{\beta} \frac{\lambda_n}{\kappa} \Psi(C_r) \ . \tag{9}$$

The above result is deterministic, but contains $\lambda_n$ and $\kappa$. In Section 3, we give precise characterizations of $\lambda_n$, which needs to satisfy (2). In Sections 4 and 5, we characterize the RSC condition constant $\kappa$ for different losses and a variety of design matrices.

# 3 Bounds on the Regularization Parameter

Recall that the parameter $\lambda_n$ needs to satisfy the inequality

$$\lambda_n \geq \beta R^*(\nabla \mathcal{L}(\theta^*; Z^n)) \,. \tag{10}$$

The right hand side of the inequality has two issues: it depends on $\theta^*$, and it is a random variable, since it depends on $Z^n$. In this section, we characterize $E[R^*(\nabla \mathcal{L}(\theta^*; Z^n))]$ in terms of the Gaussian width of the unit norm ball $\Omega_R = \{u : R(u) \leq 1\}$, and also discuss large deviation bounds around the expectation. For ease of exposition, we present results for the case of squared loss, i.e., $\mathcal{L}(\theta^*; Z^n) = \frac{1}{2n}\|y - X\theta^*\|^2$ with the linear model $y = X\theta + \omega$, where $\omega$ can be Gaussian or sub-Gaussian noise. For this setting, $\nabla \mathcal{L}(\theta^*; Z^n) = \frac{1}{n}X^T(y - X\theta^*) = \frac{1}{n}X^T\omega$. The analysis can be extended to GLMs, using analysis techniques discussed in Section 5.

**Gaussian Designs:** First, we consider Gaussian design $X$, where $x_{ij} \sim N(0, 1)$ are independent, and $\omega$ is elementwise independent Gaussian or sub-Gaussian noise.

**Theorem 3** *Let $\Omega_R = \{u : R(u) \leq 1\}$. Then, for Gaussian design $X$ and Gaussian or sub-Gaussian noise $\omega$, for a suitable constant $\eta_0 > 0$, we have*

$$E[R^*(\nabla \mathcal{L}(\theta^*; Z^n))] \leq \frac{\eta_0}{\sqrt{n}} w(\Omega_R) \,. \tag{11}$$

*Further, for any $\tau > 0$, for suitable constants $\eta_1, \eta_2 > 0$, with probability at least $(1 - \eta_1 \exp(-\eta_2 \tau^2))$*

$$R^*(\nabla \mathcal{L}(\theta^*; Z^n)) \leq \frac{\eta_0}{\sqrt{n}} w(\Omega_R) + \frac{\tau}{\sqrt{n}} \,. \tag{12}$$

For anisotropic Gaussian design, i.e., when columns of $X \in \mathbb{R}^{n \times p}$ have covariance $\Sigma_{p \times p}$, the above result continues to hold with $w(\Omega_R)$ replaced by $\sqrt{\Lambda_{\max}(\Sigma)} w(\Omega_R)$, where $\Lambda_{\max}(\Sigma)$ denotes the operator norm (largest eigenvalue). For correlated isotropic design, i.e., when rows of $X \in \mathbb{R}^n$ have covariance $\Gamma_{n \times n}$, the result continues to hold with $w(\Omega_R)$ replaced by $\sqrt{\Lambda_{\max}(\Gamma)} w(\Omega_R)$.

**Sub-Gaussian Designs:** Recall that for a sub-Gaussian variable $x$, the sub-Gaussian norm $\|x\|_{\psi_2} = \sup_{p \geq 1} \frac{1}{\sqrt{p}}(E[|x|^p])^{1/p}$ [18]. Now, we consider sub-Gaussian design $X$, where $\|x_{ij}\|_{\psi_2} \leq k$ and $x_{ij}$ are i.i.d., and $\omega$ is elementwise independent Gaussian or sub-Gaussian noise.

**Theorem 4** *Let $\Omega_R = \{u : R(u) \leq 1\}$. Then, for sub-Gaussian design $X$ and Gaussian or sub-Gaussian noise $\omega$, for a suitable constant $\eta_0 > 0$, we have*

$$E[R^*(\nabla \mathcal{L}(\theta^*; Z^n))] \leq \frac{\eta_0}{\sqrt{n}} w(\Omega_R) \,. \tag{13}$$

Interestingly, the analysis for the result above involves 'sub-Gaussian width' which can be upper bounded by a constant times the Gaussian width, using generic chaining [15]. Further, one can get Gaussian-like exponential concentration around the expectation for important classes of sub-Gaussian random variables, including bounded random variables [6], and when $X_u = \langle h, u \rangle$, where $u$ is any unit vector, are such that their Malliavin derivatives have almost surely bounded norm in $L^2[0, 1]$, i.e., $\int_0^1 |D_r X_u|^2 dr \leq \eta$ [19].

Next, we provide a mechanism for bounding the Gaussian width $w(\Omega_R)$ of the unit norm ball in terms of the Gaussian width of a suitable cone, obtained by shifting or translating the norm ball. In particular, the result involves taking any point on the boundary of the unit norm ball, considering that as the origin, and constructing a cone using the norm ball. Since such a construction can be done with any point on the boundary, the tightest bound is obtained by taking the infimum over all points on the boundary. The motivation behind getting an upper bound of the Gaussian width $w(\Omega_R)$ of the unit norm ball in terms of the Gaussian width of such a cone is because considerable advances have been made in recent years in upper bounding Gaussian widths of such cones.

**Lemma 3** *Let $\Omega_R = \{u : R(u) \leq 1\}$ be the unit norm ball and $\Theta_R = \{u : R(u) = 1\}$ be the boundary. For any $\tilde{\theta} \in \Theta_R$, $\rho(\tilde{\theta}) = \sup_{\theta:R(\theta) \leq 1} \|\theta - \tilde{\theta}\|_2$ is the diameter of $\Omega_R$ measured with respect to $\tilde{\theta}$. Let $G(\tilde{\theta}) = \text{cone}(\Omega_R - \tilde{\theta}) \cap \rho(\tilde{\theta}) B_2^p$, i.e., the cone of $(\Omega_R - \tilde{\theta})$ intersecting the ball of radius $\rho(\tilde{\theta})$. Then*

$$w(\Omega_R) \leq \inf_{\tilde{\theta} \in \Theta_R} w(G(\tilde{\theta})) \,. \tag{14}$$

# 4 Least Squares Models: Restricted Eigenvalue Conditions

When the loss function is squared loss, i.e., $\mathcal{L}(\theta; Z^n) = \frac{1}{2n}\|\mathbf{y} - X\theta\|^2$, the RSC condition (6) becomes equivalent to the Restricted Eigenvalue (RE) condition [2, 9], i.e., $\frac{1}{n}\|X\Delta\|_2^2 \geq \kappa\|\Delta\|_2^2$, or equivalently, $\frac{\|X\Delta\|_2}{\|\Delta\|_2} \geq \sqrt{\kappa n}$ for any $\Delta$ in the error cone $C_r$. Since the absolute magnitude of $\|\Delta\|_2$ does not play a role in the RE condition, without loss of generality we work with unit vectors $u \in A = C_r \cap S^{p-1}$, where $S^{p-1}$ is the unit sphere.

In this section, we establish RE conditions for a variety of Gaussian and sub-Gaussian design matrices, with isotropic, anisotropic, or dependent rows, i.e., when samples (rows of $X$) are correlated. Results for certain types of design matrices for certain types of norms, especially the $L_1$ norm, have appeared in the literature [2, 13, 14]. Our analysis considers a wider variety of design matrices and establishes RSC conditions for any $A \subseteq S^{p-1}$, thus corresponding to any norm. Interestingly, the Gaussian width $w(A)$ of $A$ shows up in all bounds, as a geometric measure of the size of the set $A$, even for sub-Gaussian design matrices. In fact, all existing RE results do implicitly have the width term, but in a form specific to the chosen norm [13, 14]. The analysis on atomic norm in [4] has the $w(A)$ term explicitly, but the analysis relies on Gordon's inequality [5, 7], which is applicable only for isotropic Gaussian design matrices.

The proof technique we use is simple, a standard covering argument, and is largely the same across all the cases considered. A unique aspect of our analysis, used in all the proofs, is a way to go from covering numbers of $A$ to the Gaussian width of $A$ using the Sudakov-Dudley inequality [7]. Our general techniques are in sharp contrast to much of the existing literature on RE conditions, which commonly use specialized tools such as Gaussian comparison principles [13, 9], and/or specialized analysis geared to a particular norm such as $L_1$ [14].

## 4.1 Restricted Eigenvalue Conditions: Gaussian Designs

In this section, we focus on the case of Gaussian design matrices $X \in \mathbb{R}^{n \times p}$, and consider three settings: (i) independent-isotropic, where the entries are elementwise independent, (ii) independent-anisotropic, where rows $X_i$ are independent but each row has a covariance $E[X_i X_i^T] = \Sigma \in \mathbb{R}^{p \times p}$, and (iii) dependent-isotropic, where the rows are isotropic but the columns $X_j$ are correlated with $E[X_j X_j^T] = \Gamma \in \mathbb{R}^{n \times n}$. For convenience, we assume $E[x_{ij}^2] = 1$, noting that the analysis easily extends to the general case of $E[x_{ij}^2] = \sigma^2$.

**Independent Isotropic Gaussian (IIG) Designs:** The IIG setting has been extensively studied in the literature [3, 9]. As discussed in the recent work on atomic norms [4], one can use Gordon's inequality [5, 7] to get RE conditions for the IIG setting. Our goal in this section is two-fold: first, we present the RE conditions obtained using our simple proof technique, and show that it is equivalent, up to constants, the RE condition obtained using Gordon's inequality, an arguably heavy-duty technique only applicable to the IIG setting; and second, we go over some facets of how we present the results, which will apply to all subsequent RE-style results as well as give a way to plug-in $\kappa$ in the estimation error bound in (9).

**Theorem 5** *Let the design matrix $X \in \mathbb{R}^{n \times p}$ be elementwise independent and normal, i.e., $x_{ij} \sim N(0,1)$. Then, for any $A \subseteq S^{p-1}$, any $n \geq 2$, and any $\tau > 0$, with probability at least $(1 - \eta_1 \exp(-\eta_2 \tau^2))$, we have*

$$\inf_{u \in A} \|Xu\|_2 \geq \frac{1}{2}\sqrt{n} - \eta_0 w(A) - \tau \,, \tag{15}$$

*$\eta_0, \eta_1, \eta_2 > 0$ are absolute constants.*

We consider the equivalent result one could obtain by directly using Gordon's inequality [5, 7]:

**Theorem 6** *Let the design matrix $X$ be elementwise independent and normal, i.e., $x_{ij} \sim N(0,1)$. Then, for any $A \subseteq S^{p-1}$ and any $\tau > 0$, with probability at least $(1 - 2\exp(-\tau^2/2))$, we have*

$$\inf_{u \in A} \|Xu\|_2 \geq \gamma_n - w(A) - \tau \,, \tag{16}$$

*where $\gamma_n = E[\|h\|_2] > \frac{n}{\sqrt{n+1}}$ is the expected length of a Gaussian random vector in $\mathbb{R}^n$.*

Interestingly, the results are equivalent, up to constants. However, unlike Gordon's inequality, our proof technique generalizes to all the other design matrices considered in the sequel.

We emphasize three additional aspects in the context of the above analysis, which will continue to hold for all the subsequent results but will not be discussed explicitly. First, to get a form of the result which can be used as $\kappa$ and plugged in to the estimation error bound (9), one can simply choose $\tau = \frac{1}{2}(\frac{1}{2}\sqrt{n} - \eta_0 w(A))$ so as to get

$$\inf_{u \in A} \|Xu\|_2 \geq \frac{1}{4}\sqrt{n} - \frac{\eta_0}{2} w(A) \ , \tag{17}$$

with high probability. Table 1 shows a summary of recovery bounds on Independent Isotropic Gaussian design matrices with Gaussian noise. Second, the result does not depend on the fact that $u \in A \subseteq C_r \cap S^{p-1}$ so that $\|u\|_2 = 1$. For example, one can consider the cone $C_r$ to be intersecting with a sphere $\rho S^{p-1}$ of a different radius $\rho$, to give $A_\rho = C_r \cap \rho S^{p-1}$ so that $u \in A_\rho$ has $\|u\|_2 = \rho$. For simplicity, let $A = A_1$, i.e., corresponding to $\rho = 1$. Then, a straightforward extension yields $\inf_{u \in A_\rho} \|Xu\|_2 \geq (\frac{1}{2}\sqrt{n} - \eta_0 w(A) - \tau)\|u\|_2$, with probability at least $(1 - \eta_1 \exp(-\eta_2 \tau^2))$, since $\|Xu\|_2 = \|X\frac{u}{\|u\|_2}\|_2 \|u\|_2$ and $w(A_{\|u\|_2}) = \|u\|_2 w(A)$ [4]. Such a scale independence is in fact necessary for the error bound analysis in Section 2. Finally, note that the leading constant $\frac{1}{2}$ was a consequence of our choice of $\epsilon = \frac{1}{4}$ for the $\epsilon$-net covering of $A$ in the proof. One can get other constants, less than 1, with different choices of $\epsilon$, and the constants $\eta_0, \eta_1, \eta_2$ will change based on this choice.

**Independent Anisotropic Gaussian (IAG) Designs:** We consider a setting where the rows $X_i$ of the design matrix are independent, but each row is sampled from an anisotropic Gaussian distribution, i.e., $X_i \sim N(0, \Sigma_{p \times p})$ where $X_i \in \mathbb{R}^p$. The setting has been considered in the literature [13] for the special case of $L_1$ norms, and sharp results have been established using Gaussian comparison techniques [7]. We show that equivalent results can be obtained by our simple technique, which does not rely on Gaussian comparisons [7, 9].

**Theorem 7** *Let the design matrix $X$ be row wise independent and each row $X_i \sim N(0, \Sigma_{p \times p})$. Then, for any $A \subseteq S^{p-1}$ and any $\tau > 0$, with probability at least $1 - \eta_1 \exp(-\eta_2 \tau^2)$, we have*

$$\inf_{u \in A} \|Xu\|_2 \geq \frac{1}{2}\sqrt{\nu}\sqrt{n} - \eta_0 \sqrt{\Lambda_{\max}(\Sigma)}\, w(A) - \tau \ , \tag{18}$$

*where $\sqrt{\nu} = \inf_{u \in A} \|\Sigma^{1/2}u\|_2$, $\sqrt{\Lambda_{\max}(\Sigma)}$ denotes the largest eigenvalue of $\Sigma^{1/2}$ and $\eta_0, \eta_1, \eta_2 > 0$ are constants.*

A comparison with the results of [13] is instructive. The leading term $\sqrt{\nu}$ appears in [13] as well—we have simply considered $\inf_{u \in A}$ on both sides, and the result in [13] is for any $u$ with the $\|\Sigma^{1/2}u\|_2$ term. The second term in [13] depends on the largest entry in the diagonal of $\Sigma$, $\sqrt{\log p}$, and $\|u\|_1$. These terms are a consequence of the special case analysis for $L_1$ norm. In contrast, we consider the general case and simply get the scaled Gaussian width term $\sqrt{\Lambda_{\max}(\Sigma)}\, w(A)$.

**Dependent Isotropic Gaussian (DIG) Designs:** We now consider a setting where the rows of the design matrix $\tilde{X}$ are isotropic Gaussians, but the columns $\tilde{X}_j$ are correlated with $E[\tilde{X}_j \tilde{X}_j^T] = \Gamma \in \mathbb{R}^{n \times n}$. Interestingly, correlation structure over the columns make the samples dependent, a scenario which has not yet been widely studied in the literature [22, 10]. We show that our simple technique continues to work in this scenario and gives a rather intuitive result.

**Theorem 8** *Let $\tilde{X} \in \mathbb{R}^{n \times p}$ be a matrix whose rows $\tilde{X}_i$ are isotropic Gaussian random vectors in $\mathbb{R}^p$ and the columns $\tilde{X}_j$ are correlated with $E[\tilde{X}_j \tilde{X}_j^T] = \Gamma$. Then, for any set $A \subseteq S^{p-1}$ and any $\tau > 0$, with probability at least $(1 - \eta_1 \exp(-\eta_2 \tau^2))$, we have*

$$\inf_{u \in A} \|\tilde{X}u\|_2 \geq \frac{3}{4}\sqrt{\operatorname{Tr}(\Gamma)} - \sqrt{\Lambda_{\max}(\Gamma)}\left(\eta_0 w(A) + \frac{5}{2}\right) - \tau \tag{19}$$

*where $\eta_0, \eta_1, \eta_2 > 0$ are constants.*

Note that with the assumption that $E[x_{ij}^2] = 1$, $\Gamma$ will be a correlation matrix implying $\operatorname{Tr}(\Gamma) = n$, and making the sample size dependence explicit. Intuitively, due to sample correlations, $n$ samples are effectively equivalent to $\frac{\operatorname{Tr}(\Gamma)}{\Lambda_{\max}(\Gamma)} = \frac{n}{\Lambda_{\max}(\Gamma)}$ samples.

## 4.2 Restricted Eigenvalue Conditions: Sub-Gaussian Designs

In this section, we focus on the case of sub-Gaussian design matrices $X \in \mathbb{R}^{n \times p}$, and consider three settings: (i) independent-isotropic, where the rows are independent and isotropic, (ii) independent-anisotropic, where the rows $X_i$ are independent but each row has a covariance $E[X_i X_i^T] = \Sigma_{\mathbf{p} \times p}$, and (iii) dependent-isotropic, where the rows are isotropic and the columns $X_j$ are correlated with $E[X_j X_j^T] = \Gamma_{n \times n}$. For convenience, we assume $E[x_{ij}^2] = 1$ and the sub-Gaussian norm $\|x_{ij}\|_{\psi_2} \le k$ [18]. In recent work, [17] also considers generalizations of RE conditions to sub-Gaussian designs, although our proof techniques are different.

**Independent Isotropic Sub-Gaussian Designs:** We start with the setting where the sub-Gaussian design matrix $X \in \mathbb{R}^{n \times p}$ has independent rows $X_i$ and each row is isotropic.

**Theorem 9** *Let $X \in \mathbb{R}^{n \times p}$ be a design matrix whose rows $X_i$ are independent isotropic sub-Gaussian random vectors in $R^p$. Then, for any set $A \subseteq S^{p-1}$ and any $\tau > 0$, with probability at least $(1 - 2\exp(-\eta_1 \tau^2))$, we have*

$$\inf_{u \in A} \|Xu\|_2 \ge \sqrt{n} - \eta_0 w(A) - \tau , \qquad (20)$$

*where $\eta_0, \eta_1 > 0$ are constants which depend only on the sub-Gaussian norm $\|x_{ij}\|_{\psi_2} = k$.*

**Independent Anisotropic Sub-Gaussian Designs:** We consider a setting where the rows $X_i$ of the design matrix are independent, but each row is sampled from an anisotropic sub-Gaussian distribution, i.e., $\|x_{ij}\|_{\psi_2} = k$ and $E[X_i X_i^T] = \Sigma_{p \times p}$.

**Theorem 10** *Let the sub-Gaussian design matrix $X$ be row wise independent, and each row has $E[X_i X_i^T] = \Sigma \in \mathbb{R}^{p \times p}$. Then, for any $A \subseteq S^{p-1}$ and any $\tau > 0$, with probability at least $(1 - 2\exp(-\eta_1 \tau^2))$, we have*

$$\inf_{u \in A} \|Xu\|_2 \ge \sqrt{\nu}\sqrt{n} - \eta_0 \Lambda_{\max}(\Sigma) w(A) - \tau , \qquad (21)$$

*where $\sqrt{\nu} = \inf_{u \in A} \|\Sigma^{1/2} u\|_2$, $\sqrt{\Lambda_{\max}(\Sigma)}$ denotes the largest eigenvalue of $\Sigma^{1/2}$, and $\eta_0, \eta_1 > 0$ are constants which depend on the sub-Gaussian norm $\|x_{ij}\|_{\psi_2} = k$.*

Note that [14] establish RE conditions for anisotropic sub-Gaussian designs for the special case of $L_1$ norm. In contrast, our results are general and in terms of the Gaussian width $w(A)$.

**Dependent Isotropic Sub-Gaussian Designs:** We consider the setting where the sub-Gaussian design matrix $\tilde{X}$ has isotropic sub-Gaussian rows, but the columns $\tilde{X}_j$ are correlated with $E[\tilde{X}_j \tilde{X}_j^T] = \Gamma$, implying dependent samples.

**Theorem 11** *Let $\tilde{X} \in \mathbb{R}^{n \times p}$ be a sub-Gaussian design matrix with isotropic rows and correlated columns with $E[\tilde{X}_j \tilde{X}_j^T] = \Gamma \in \mathbb{R}^{n \times n}$. Then, for any $A \subseteq S^{p-1}$ and any $\tau > 0$, with probability at least $(1 - 2\exp(-\eta_1 \tau^2))$, we have*

$$\inf_{u \in A} \|\tilde{X}u\|_2 \ge \frac{3}{4}\sqrt{\mathrm{Tr}(\Gamma)} - \eta_0 \Lambda_{\max}(\Gamma) w(A) - \tau , \qquad (22)$$

*where $\eta_0, \eta_1$ are constants which depend on the sub-Gaussian norm $\|x_{ij}\|_{\psi_2} = k$.*

## 5 Generalized Linear Models: Restricted Strong Convexity

In this section, we consider the setting where the conditional probabilistic distribution of $y|x$ follows an exponential family distribution: $p(y|x; \theta) = \exp\{y\langle\theta, x\rangle - \psi(\langle\theta, x\rangle)\}$, where $\psi(\cdot)$ is the log-partition function. Generalized linear models consider the negative likelihood of such conditional distributions as the loss function: $\mathcal{L}(\theta; Z^n) = \frac{1}{n}\sum_{i=1}^{n}(\psi(\langle\theta, X_i\rangle) - \langle\theta, y_i X_i\rangle)$. Least squares regression and logistic regression are popular special cases of GLMs. Since $\nabla\psi(\langle\theta, x\rangle) = E[y|x]$, we have $\nabla\mathcal{L}(\theta^*; Z^n) = \frac{1}{n}X^T\omega$, where $\omega_i = \nabla\psi(\langle\theta, X_i\rangle) - y_i = E[y|X_i] - y_i$ plays the role of noise. Hence, the analysis in Section 3 can be applied assuming $\omega$ is Gaussian or sub-Gaussian. To obtain RSC conditions for GLMs, first note that

$$\delta\mathcal{L}(\theta^*, \Delta; Z^n) = \frac{1}{n}\sum_{i=1}^{n} \nabla^2\psi(\langle\theta^*, X_i\rangle + \gamma_i\langle\Delta, X_i\rangle)\langle\Delta, X_i\rangle^2 , \qquad (23)$$

*Table 1:* A summary of various values for $L_1$ and $L_\infty$ norms with all values correct upto constants.

| $R(u)$ | $\lambda_n := c_1 \frac{w(\Omega_R)}{\sqrt{n}}$ | $\kappa := \left[\max\left\{\left(1 - c_2 \frac{w(A)}{\sqrt{n}}\right), 0\right\}\right]^2$ | $\Psi(C_r)$ | $\|\hat{\Delta}_n\|_2 := c_3 \frac{\Psi(C_r)\lambda_n}{\kappa}$ |
|---|---|---|---|---|
| $\ell_1$ norm | $O\left(\sqrt{\frac{\log p}{n}}\right)$ | $O(1)$ | $\sqrt{s}$ | $O\left(\sqrt{\frac{s \log p}{n}}\right)$ |
| $\ell_\infty$ norm | $O\left(\sqrt{\frac{p}{2n}}\right)$ | $O(1)$ | $1$ | $O\left(\sqrt{\frac{p}{2n}}\right)$ |

where $\gamma_i \in [0, 1]$, by mean value theorem. Since $\psi$ is of Legendre type, the second derivative $\nabla^2 \psi(\cdot)$ is always positive. Since the RSC condition relies on a non-trivial lower bound for the above quantity, the analysis considers a suitable compact set where $\ell = \ell_\psi(T) = \min_{|a| \leq 2T} \nabla^2 \psi(a)$ is bounded away from zero. Outside this compact set, we will only use $\nabla^2 \psi(\cdot) > 0$. Then,

$$\delta\mathcal{L}(\theta^*, \Delta; Z^n) \geq \frac{\ell}{n} \sum_{i=1}^n \langle X_i, \Delta \rangle^2 \, \mathbb{I}[|\langle X_i, \theta^* \rangle| < T] \, \mathbb{I}[|\langle X_i, \Delta \rangle| < T] . \tag{24}$$

We give a characterization of the RSC condition for independent isotropic sub-Gaussian design matrices $X \in \mathbb{R}^{n \times p}$. The analysis can be suitably generalized to the other design matrices considered in Section 4 by using the same techniques. As before, we denote $\Delta$ as $u$, and consider $u \in A \subseteq S^{p-1}$ so that $\|u\|_2 = 1$. Further, we assume $\|\theta^*\|_2 \leq c_1$ for some constant $c_1$. Assuming $X$ has sub-Gaussian entries with $\|x_{ij}\|_{\psi_2} \leq k$, $\langle X_i, \theta^* \rangle$ and $\langle X_i, u \rangle$ are sub-Gaussian random variables with sub-Gaussian norm at most $Ck$. Let $\phi_1 = \phi_1(T; u) = P\{|\langle X_i, u \rangle| > T\} \leq e \cdot \exp(-c_2 T^2 / C^2 k^2)$, and $\phi_2 = \phi_2(T; \theta^*) = P\{|\langle X_i, \theta^* \rangle| > T\} \leq e \cdot \exp(-c_2 T^2 / C^2 k^2)$. The result we present is in terms of the constants $\ell = \ell_\psi(T)$, $\phi_1 = \phi(T; u)$ and $\phi_2 = \phi(T, \theta^*)$ for any suitably chosen $T$.

**Theorem 12** *Let $X \in \mathbb{R}^{n \times p}$ be a design matrix with independent isotropic sub-Gaussian rows. Then, for any set $A \subseteq S^{p-1}$, any $\alpha \in (0,1)$, any $\tau > 0$, and any $n \geq \frac{2}{\alpha^2(1-\phi_1-\phi_2)}(cw^2(A) + \frac{c_3(1-\phi_1-\phi_2)^5}{c_4^4 k^4}(1-\alpha)\tau^2)$ for suitable constants $c_3$ and $c_4$, with probability at least $1 - 3\exp\left(-\eta_1 \tau^2\right)$, we have*

$$\inf_{u \in A} \sqrt{n \partial \mathcal{L}(\theta^*; u, X)} \geq \ell\sqrt{\pi}\left(\sqrt{n} - \eta_0 w(A) - \tau\right), \tag{25}$$

*where $\pi = (1-\alpha)(1-\phi_1-\phi_2)$, $\ell = \ell_\psi(T) = \min_{|a| \leq 2T+K} \nabla^2 \psi(a)$, and constants $(\eta_0, \eta_1)$ depend on the sub-Gaussian norm $\|x_{ij}\|_{\psi_2} = k$.*

The form of the result is closely related to the corresponding result for the RE condition on $\inf_{u \in A} \|Xu\|_2$ in Section 4.2. Note that RSC analysis for GLMs was considered in [9] for specific norms, especially $L_1$, whereas our analysis applies to any set $A \subseteq S^{p-1}$, and hence to any norm. Further, following similar argument structure as in Section 4.2, the analysis for GLMs can be extended to anisotropic and dependent design matrices.

## 6 Conclusions

The paper presents a general set of results and tools for characterizing non-asymptotic estimation error in norm regularized regression problems. The analysis holds for any norm, and includes much of existing literature focused on structured sparsity and related themes as special cases. The work can be viewed as a direct generalization of results in [9], which presented related results for decomposable norms. Our analysis illustrates the important role Gaussian widths, as a geometric measure of size of suitable sets, play in such results. Further, the error sets of regularized and constrained versions of such problems are shown to be closely related [2]. Going forward, it will be interesting to explore similar generalizations for the semi-parametric and non-parametric settings.

**Acknowledgements:** We thank the anonymous reviewers for helpful comments and suggestions on related work. We thank Sergey Bobkov, Snigdhansu Chatterjee, and Pradeep Ravikumar for discussions related to the paper. The research was supported by NSF grants IIS-1447566, IIS-1422557, CCF-1451986, CNS-1314560, IIS-0953274, IIS-1029711, and by NASA grant NNX12AQ39A.

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
