[Reviews · NeurIPS 2014]

Submitted by Assigned_Reviewer_5

This paper studies the structured estimation problem where the goal is to find a structured parameter based on regularized regression. The paper presents a general framework, giving results for any norm regularizer, least squares and GLM-based losses, three types of design matrices, and two types of noise.

This paper builds on a series of papers on structured estimation problems. This paper gives a very general analysis of the regularized formulation of these problems, a topic which has yet to be studied in such generality. That being said, the constrained version has been studied quite extensively. The connections between the error and the gaussian width of the cone induced by the regularizer are by now well known [6], and already do not require anything except convexity of the regularizer. The analysis of the different noise models also seems to be considered by [6] and several other papers that simply consider noise of bounded magnitude. This paper builds on these connections by establishing them for the regularized version of these problems.

The paper also uses existing measure concentration tools to consider various design matrices (gaussian/subgaussian with correlated rows and columns). While these tools are fairly standard, they have not been applied to the structured estimation setting.
Summary: Many of the pieces in this paper are by fairly well known to our community, but this paper contributes by putting the pieces together to yield interesting results. Since many pieces are well known, the results are not particularly surprising, but I do think deriving all of the results is a substantial contribution.

Submitted by Assigned_Reviewer_19

1) Please give examples of what new regularizers are included in your analysis. What do you cover that is not decomposable, yet yields fast estimation rates?
2) You give several generalizations of Gordon's inequality to correlated Gaussian and subGaussian design. These are only useful if we can compute the Gaussian width of the set. The few sets that we do know the Gaussian width for are sparse vectors or the similar error cone from Bickel et al. There are several examples in [6] that aren't decomposable and you may want to emphasize these norms, since they have known gaussian widths and aren't decomposable.
3) Without specific examples of error cones, and their Gaussian widths, and the regularizers, the importance of the results here are not clear. For example, the regularizers covered in this work are the same as those in [6], which is broader than those in Negahban et al. Do you obtain the same rates of convergence for the Lagrangian formulation as the constrained formulation? This is expected, but it would be nice to see a few examples.
4) I just came across a paper http://arxiv.org/pdf/1405.1102.pdf , and I think in Theorem 6.3 on page 11 it contains essentially the same results for "Independent anistotropic sub-gaussian designs". The theorem is stated for sets that are cone intersected with sphere, but of course any closed subset of the sphere can be represented like that. Thus it also applies to all subsets of the sphere. It may not be fair to compare against this paper since it was posted only a month before the NIPS submission deadline. On the other hand, the proof in the Tropp paper seems really easy. Can you please comment?

The paper by Mendelson et al. 2007 is also relevant, http://link.springer.com/article/10.1007%2Fs00039-007-0618-7
Please discuss the relationship to these two papers.
Summary: Without specific examples of error cones, and their Gaussian widths, and the regularizers, the importance of the results here are not clear. A bit of discussion on how the results in this paper fit into existing literature, and which results are new would considerably improve this work. I believe this work is technically impressive, and it is very nice to have such a collection of results all in one paper. After reading the author response, I am satisfied that this paper has contributed new results to existing literature.

Submitted by Assigned_Reviewer_24

This is a very well-written paper providing fairly comprehensive and interesting extensions to a growing body of work on norm regularized regression. The authors generalize the already general work of Negahban et al. '12 and shed light on RSC-like conditions defined in terms of Gaussian widths of error cones. Their methods involve a nice blend of random matrix theory and other high-dimensional concentration techniques.

The contribution of the paper already seems very significant, and the proof techniques will no doubt be useful in future work. However, some suggestions for improvement are the following:

(1) the applicability of Lemma 2 -- when is the l_2-norm bound actually useful, and are there cases where the rate obtained from Lemma 2 is actually optimal?

and

(2) the computation of the compatibility constant Psi(C_r) -- what scaling might one expect over various error cones of interest? Perhaps some examples would be helpful.

Also, there seems to be some notational confusion in Section 2.2 concerning E_r versus C_r, which the authors should correct in the revision.
Summary: This is a very nice paper with clear theoretical contributions that will be of interest to the NIPS community.
Author Feedback
Author rebuttal: Reviewer_19:

Thanks for the comments and pointers.

1) The recently introduced k-support norm is a good example of a non-decomposable norm with known Gaussian width, where our analysis yields fast rates. We can add this as an example in the paper.

2) Since our analysis applies to all norms, one can directly instantiate the analysis for atomic norms with known Gaussian widths as discussed in [6]. We will add a remark with some examples.

3) Yes, for the regularized setting, we do obtain same rates as the constrained case as in [6]. This is indeed reassuring. An equivalent result was established for the L1 case - Lasso (regularized) vs Dantzig (constrained) – by Bickel et al., [2]. Our analysis applies to all norms. We will give examples to illustrate the point.

4) Thanks for the pointers. They are indeed relevant, and Theorem 6.3 in Tropp’s recent work [T14] is related to our IAS case. Interestingly, the proof techniques appear different. We use a simple covering argument for all the six cases we consider, along with the use of the so-called `weak converse’ of Dudley’s inequality. [T14] builds on results in [KM13] and [MPTJ07] in his paper. There may be value to having different proof techniques. Further, our approach generalizes to the dependent sample case, for which there are no existing results, e.g., in [T14], [6], etc.

In terms of timing w.r.t. [T14], we had our results ready in May – but did not submit to arXiv, since we were planning to submit this to NIPS. We are hoping this will not be held against our work.

Reviewer_24:

Thanks for the encouraging comments and suggestions.
As Reviewer_24 correctly points out, our results directly generalize the regularized framework of Negahban et al., [12]. The results in [12] apply only to decomposable norms, whereas our analysis applies to all norms. Further, we present our results in terms of Gaussian widths of sets, which makes it possible to compare the results with the constrained counterparts, e.g., as in [6].

1) For Lemma 2, we do not have a lower bound analysis, so it is hard to say if the rates are optimal for general norms. For the widely studied examples, such as L1, the rates match the existing results, up to constants.
It is of interest to characterize other norms of the error vector. We will explore this in future work.

2) For the compatibility constant \Psi, we will add some examples, e.g., for L1, the constant is \sqrt{s} if the optimal vector is s-sparse. [12] also has a discussion on this constant.

There is indeed a notational confusion between E_r and C_r. We have found a few other typos, which will all be fixed in the final version.

Reviewer_5:

The reviewer raises concerns regarding the novelty of the work, which we try to address below.

While there has indeed been significant amount of work on related themes, there are still several big gaps in the literature, and our paper addresses a set of these open questions.

As an example, consider the following problem: \min_\theta ||y – X \theta ||^2 + \lambda R(\theta), where R(.) is say the k-support norm (hence non-decomposable), and the design matrix X has sub-Gaussian and possibly dependent rows. We do not know of any existing paper which can characterize the recovery guarantees for the above example problem. The original k-support norm paper (NIPS’12) leaves the statistical analysis as an open question. The above is a special case of our analysis, and one can readily get recovery guarantees using our results.

[6] indeed covers a lot of ground but (i) the analysis is for a constrained problem, and (ii) they assume the design matrix X is Gaussian. In fact, their analysis techniques (e.g., Gordon’s inequality) are specific to Gaussians, and have no direct counter-part for sub-Gaussian designs.

[12] is closest in spirit to our work in that they also consider the regularized case. However, their results are applicable only to decomposable norms, which exclude several widely used norms, such as those in overlapping group Lasso, k-support norm, infimum over quadratics norms, etc. Further, unlike the analysis in [6], Gaussian widths of error cones do not show up explicitly in the regularized analysis for [12], which makes a direct comparison of the rates for the regularized and constrained cases difficult. Note that [2] compares the results for the regularized and constrained case for the L1 norm, but such analysis has not been extended to other norms.

Our work (i) generalizes the analysis of [12] to include all norms, (ii) characterizes the results in terms of Gaussian widths, so that the regularized and constrained case results can be directly compared, e.g., with results in [6], (iii) give a characterization of the regularization parameter \lambda also in terms of Gaussian width of a suitable set, (iv) considers sub-Gaussian as well as dependent design matrices (as far as we know, the dependent case has not been analyzed even for Lasso), (v) extends the results to GLMs, among other results. We do borrow some techniques from the existing literature, such as the excellent survey in [19], but there is novelty in how we use such existing results, e.g., a covering argument along with weak-converse of Dudley’s inequality has not been considered before for the analysis of norm regularized regression.